# Morphological and Transcriptome Analysis of the Near-Threatened Orchid *Habenaria radiata* with Petals Shaped Like a Flying White Bird

**DOI:** 10.3390/plants14030393

**Published:** 2025-01-28

**Authors:** Seiji Takeda, Yuki Nishikawa, Tsutomu Tachibana, Takumi Higaki, Tomoaki Sakamoto, Seisuke Kimura

**Affiliations:** 1Graduate School of Life and Environmental Sciences, Kyoto Prefectural University, Shimogamo-Hangi-Cho, Sakyo-ku, Kyoto 606-8522, Japan; 2Biotechnology Research Department, Kyoto Prefectural Agriculture Forestry and Fisheries Technology Center, Kitainayaduma-Oji 74, Seika-Cho, Kyoto 619-0244, Japan; 3International Research Center for Agricultural and Environmental Biology (IRCAEB), 2-39-1 Kurokami, Chuo-ku, Kumamoto 860-8555, Japan; 4Faculty of Advanced Science and Technology, Kumamoto University, 2-39-1 Kurokami, Chuo-ku, Kumamoto 860-8555, Japan; 5International Research Organization for Advanced Science and Technology (IROAST), Kumamoto University, 2-39-1 Kurokami, Chuo-ku, Kumamoto 860-8555, Japan; 6Center for Plant Sciences, Kyoto Sangyo University, Kamigamo-Motoyama, Kita-Ku, Kyoto 603-8555, Japan; 7Department of Industrial Life Sciences, Faculty of Life Sciences, Kyoto Sangyo University, Kamigamo-Motoyama, Kita-Ku, Kyoto 603-8555, Japan

**Keywords:** lip (labellum), petal morphology, serration

## Abstract

Orchids have evolved flowers with unique morphologies through coevolution with pollinators, such as insects. Among the floral organs, the lip (labellum), one of the three petals, exhibits a distinctive shape and plays a crucial role in attracting pollinators and facilitating pollination in many orchids. The lip of the terrestrial orchid *Habenaria radiata* is shaped like a flying white bird and is believed to attract and provide a platform for nectar-feeding pollinators, such as hawk moths. To elucidate the mechanism of lip morphogenesis, we conducted time-lapse imaging of blooming flowers to observe the extension process of the lip and analyzed the cellular morphology during the generation of serrations. We found that the wing part of the lip folds inward in the bud and fully expands in two hours after blooming. The serrations of the lip were initially formed through cell division and later deepened through polar cell elongation. Transcriptome analysis of floral buds revealed the expression of genes involved in floral organ development, cell division, and meiosis. Additionally, genes involved in serration formation are also expressed in floral buds. This study provides insights into the mechanism underlying the formation of the unique lip morphology in *Habenaria radiata*.

## 1. Introduction

Many flowers have evolved their morphology to attract pollinators. Generally, they consist of four distinct organs: sepals, petals, stamens, and carpels, whose identities are determined by a combination of floral homeotic genes. This is known as the floral ABCE model, where each class of floral homeotic genes is labeled as A, B, C, or E, and they act together to establish the identity of each organ in almost all flowering plants [1,2,3]. Although the basic structure of flowers, such as the arrangement of organs in a whorled pattern, is remarkably conserved, the morphology and number of each organ can vary widely. A remarkable example of this variation is orchid flowers. Despite having a simple basic structure—three sepals (or outer tepals) in the first whorl, three petals (or inner tepals) in the second whorl, and a single column where male and female organs are combined, approximately 28,000 species exhibit a diverse array of unique shapes and colors, often influenced by coevolution with pollinators [4]. In particular, many orchid species develop the two lateral petals and the lip (or labellum) into distinctly different shapes, playing a crucial role in attracting pollinators.

Most of the floral homeotic genes encode MADS-box transcription factors, and they have been identified in ornamental orchids, including *Phalaenopsis* [5,6,7], *Oncidium* [8,9,10], *Dendrobium* [11], *Orchis* [12], *Erycina* [13], and *Habenaria* [14,15,16]. Two main models have been proposed to explain the morphological differences between the lateral petal and the lip. One is the “orchid code”, which suggests that the class B homeotic gene *DEFICIENS* (*DEF*), functionally differentiated into four genes by gene duplication, alters the morphology of the lateral petal and lip through differential expression in the floral bud [17,18,19,20,21]. The other is the perianth code (P-code), which proposes that, in addition to the class B genes, *AGAMOUS-LIKE 6* (*AGL6*) forms distinct protein complexes, leading to changes in petal morphology [22]. In either case, it is clear that the spatiotemporal regulation of DEF-type class B genes and AGL6, along with their protein interactions, results in differences in the identity of the second whorl organs. These models are believed to be conserved in many orchid flowers; however, the morphology of the lip varies greatly among species, suggesting that each species has its own unique lip morphogenesis process, although the mechanisms remain largely unknown.

*Habenaria* is a large genus within the Orchidaceae family, containing 898 species found in tropical and subtropical regions [23]. *Habenaria radiata* (Thunb.) Spreng., syn. *Pecteilis radiata* (Thunb.) Raf, is a terrestrial orchid native to wetlands in Japan, China, South Korea, and Far East Russia. The habitats of *Habenaria radiata* have been reduced due to environmental destruction and overexploitation, and as a result, this species is listed as Near Threatened on the Red List of Japan [24]. The leaves of *Habenaria radiata* are simple, narrow, and elongated, similar to many other orchids, while its floral organs exhibit a complex shape (Appendix A). The flower consists of three green sepals, two lateral petals, a lip (labellum), a spur that is a long tubular structure originating from the base of the lip that accumulates nectar, and a column where the male and female reproductive organs are fused (Appendix A). The lip is the most distinctive part of the floral organs, with a shape resembling a white bird in flight, consisting of a central body part and two lateral wings (Appendix A). This morphology is believed to be crucial for attracting diurnal butterflies (such as *Parnara guttata*, *Polytremis pellucida*, and *Pelopidas mathias*) and nocturnal hawk moths (such as *Theretra oldenlandiae*, *Theretra japonica*, and *Theretra nessus*) [25,26,27,28]. However, the mechanisms underlying the formation of the serrations on the edges of the petals, how the lip develops in the bud, and how it unfolds during blooming have not been fully understood.

To understand the mechanism to generate the unique shape of lip in *Habenaria radiata*, we investigated how the complex morphology of the lip of *Habenaria radiata* is formed and how it expands during blooming. Additionally, we conducted transcriptome analysis of floral buds during the early stages of serration formation and comprehensively obtained gene sequences related to floral organ development. These results provide insights into the mechanism underlying the formation of the complex morphology of orchid petals.

## 2. Results

### 2.1. Anthesis and Petal Withering

To examine lip morphogenesis during anthesis and withering, we captured time-lapse photos of flowers grown in the greenhouse using fixed-position shooting. Interval shots taken every 10 min revealed that the anthesis of a flower—i.e., from sepal opening to petal expansion—takes about 3.5 to 4 h (Figure 1A–C, Appendix A). The flowers remained open for more than a week, and we observed that the upper flower began petal withering earlier than the lower flower, despite the lower flower having opened earlier (Figure 1F,G). We found that one of the two pollinia was lost in the upper flower (Figure 1D,E), but two pollinia remained in the lower flower. This suggests that pollinia removal, likely by pollinator insects, and subsequent pollination stimulates petal withering. Since the lip shape may help attract pollinators such as moths, petals that have completed their role may quickly wither. Petal withering took approximately 8 to 9 h (Figure 1F,G, Appendix A). Focusing on the expansion of the lip, the body part opened first, followed by the unfolding of the wing part, which was initially folded inward (Figure 1H,I). The entire lip expansion process took approximately 2 h.

### 2.2. Lip Morphogenesis

To investigate lip development, lips from floral buds of different sizes (1 mm, 2 mm, 4 mm, 6 mm, and 7 mm in bud length, Figure 2A–E) were dissected and analyzed under a microscope. In the 1 mm floral bud, the lip was flat and round in shape, corresponding to the ‘body’ part (Figure 2F and Appendix A). The ‘wings’ began to form on the proximal side of the lip (Figure 2G) and elongated laterally (Figure 2H). The wings developed serrations on the outer periphery, which deepened during growth, resulting in distinct serrations in the lip (Figure 2I). Vascular tissue developed in each separate part (Figure 2J). The spur initiated at the bottom of the buds when the bud length reached 2 mm (Figure 2B) and developed into a long tubular structure (Figure 2C–E).

To examine the cellular dynamics during serration development at the lip periphery, lips from different stages were dissected, and the shape of epidermal cells at the serration region was analyzed using fluorescent dye, confocal microscopy, and imaging analysis software (Figure 3A). At early stages, the cell area did not increase; instead, the cell number increased (Figure 3B_1_–B_3_). Later, each cell increased in volume (Figure 3B_4_,B_5_). The increase in cell volume depends on the directional or polarized cell growth. At early stages, the direction of cell growth was random (Figure 3C_1_–C_3_), but later, most cells elongated along the direction of the serration (Figure 3C_4_,C_5_). In summary, lip serration begins with the local activation of cell division, followed by cell elongation that increases volume and directs growth, deepening the serration (Figure 3D_1_–D_5_).

### 2.3. Transcriptome Analysis of Floral Buds

To identify genes involved in floral organ development, particularly those related to the early formation of lip serrations, RNA was extracted from buds measuring 3 mm, 4 mm, and 5 mm in length, as well as from leaves, followed by transcriptome analysis. Approximately 7500 genes were identified as being expressed more than twice as much in the buds compared to the leaves, and 6127 of these genes were commonly expressed in all the buds (Figure 4A). Gene Ontology (GO) analysis of the common genes revealed those related to floral organ formation, mitosis, and meiosis (Appendix A). Among them, 152 genes related to floral organ formation, including floral homeotic genes, were identified (Appendix A). This analysis comprehensively revealed the transcript sequences expressed in *Habenaria radiata* flowers.

A self-organizing map (SOM) analysis was performed on genes exhibiting expression changes in the 3 mm to 5 mm buds, clustering genes with similar expression patterns (Figure 4B). Clusters with increasing expression (G, H, I) contained genes related to meiosis (Appendix A), while clusters with decreasing expression (A, B, C) did not (Appendix A), indicating that meiosis of reproductive cells is occurring in the column at this stage.

### 2.4. Expression of Floral Homeotic Genes in Floral Organs

Floral homeotic genes are crucial for floral organ development. We identified the sequences of these genes from the transcriptome data and cloned some of them, which had not been previously reported, using a PCR-based method. We identified *Habenaria radiata APETALA2* (*HrAP2*) as a class A gene, *GLOBOSA* (*HrGLO1*, *HrGLO2*) and *DEFICIENS* (*HrDEF-C1*, *HrDEF-C3*, *HrDEF-C4*) as class B genes, and *AGAMOUS* (*HrAG2*, *HrAG3*) as class C genes. Additionally, we found the *AGL6* genes (*HrAGL6-C1*, *HrAGL6-C2*, *HrAGL6-C3*), as *AGL6* is involved in petal differentiation in orchids [22]. Among these, all of the class B genes, *HrAG2*, *HrAGL6-C1*, and *HrAGL6-C2*, have been previously reported [14,15,16].

We examined the expression patterns of these genes in floral organs using RT-PCR (Figure 5). *HrAP2* was expressed in all floral organs, including the leaves. *HrGLO1* and *HrGLO2* were expressed in all floral organs, whereas the *HrDEF* genes showed differential expression. Specifically, *HrDEF-C3* was expressed in both the petals and the column, while *HrDEF-C4* was strongly expressed in the column. *HrAG2* was expressed in the column, whereas *HrAG3* was strongly expressed in lateral petals but weakly expressed in other floral organs, suggesting that *HrAG3* may have a function distinct from class C in flower development. *HrAGL6* genes were expressed in the sepals and the column, and of particular interest, they exhibited different expression patterns in the petals. *HrAGL6-C1* was either not expressed or weakly expressed in petals, *HrAGL6-C2* was expressed in the lip, and *HrAGL6-C3* was more highly expressed in lateral petals than in the lip. These expression patterns support previous reports [14,15,16] and the two orchid models explaining the differentiation of second whorl organs [17,18,19,20,21,22].

### 2.5. Transgenic Analysis of Habenaria radiata Floral Homeotic Genes in a Model Plant

To investigate whether these genes in *Habenaria radiata* function as floral homeotic genes, we overexpressed them in floral mutants of *Arabidopsis thaliana* [1,2]. The constructs were transformed to the heterozygous homeotic mutants, since they produce few or no seeds due to the transformation of floral organs. The transformants were screened on media and genotyped for each mutation. A class A mutant, *ap2-3*, develops carpels in the first whorl in place of sepals, along with fewer petals and stamens (Figure 6A). The screening resulted in 14 T1 plants: three wild-type, four heterozygous, and six homozygous. Two homozygous lines produced sepal-like organs in the first whorl and more stamens, and some stamens were petaloid in the anther region (Figure 6B). These results suggest that *HrAP2* can weakly replace *AP2* in *Arabidopsis thaliana*, indicating that it has weak activity as a class A gene.

We introduced overexpression constructs of *HrAG2* or *HrAG3* into *ag-1* heterozygous plants. The *ag-1* plants developed only sepals and petals, resulting in double flowers (Figure 6C). Fourteen T1 plants were obtained: four wild-type, six heterozygous, and four homozygous. In the homozygous background, flowers developed stamen-like organs instead of petals (Figure 6D). In wild-type and heterozygous siblings, flowers developed stamens in place of petals (Figure 6E). Most of the transformants exhibited hyponastic leaves (Figure 6E). These results suggests that *HrAG2* functions as a class C gene in flower development. In contrast, overexpression of *HrAG3* had no effect on floral organ or leaf development, suggesting that *HrAG3* may have a different function in flower development. We also overexpressed the *HrDEF-C4* in *ap3-5* mutants, but observed no effect on floral organs.

Although they are phylogenetically distant, *HrAP2* and *HrAG2* partially replaced the floral homeotic genes in *Arabidopsis thaliana*, suggesting that these genes may also function as floral homeotic genes in *Habenaria radiata*.

### 2.6. Digital Expression of Genes Involved in Serration Formation

In *Habenaria radiata* flowers, the formation of lip serrations is a key process for pollination. Therefore, we focused on genes associated with serration formation in plant organs and examined their digital gene expression (DGE) levels. Homologous genes of *CUC2*, *CUC3*, and *BZR1*, which regulate serration formation in *Arabidopsis thaliana* leaves [29,30,31], were highly expressed, while the homologous gene of *DPA4*, a negative regulator of *CUC2*, showed reduced expression (Figure 7) [31,32]. These factors are suggested to be involved in lip serration formation. On the other hand, homologous genes of *SALAD*, known negative regulators of *CUC2*, were highly expressed in floral buds (Figure 7). The *SALAD* gene has been shown to modulate the depth of serrations in strawberry leaves by repressing *CUC2* expression [33], suggesting that it may play a role in shaping serrations rather than promoting their growth. Homologous genes of *TCP2*, *TCP3*, and *TCP4*, which are known to regulate serration formation in leaves and petals negatively [34,35,36], showed increased expression in buds compared to leaves (Figure 7). Since *CINCINATA*, from the same family in *Antirrhinum*, is involved in lobe development in petals [37], these *TCP* genes are likely involved in floral organ formation in processes other than serration formation.

## 3. Discussion

To understand the mechanism behind the generation of lip serration, which characterizes the unique shape of the *Habenaria radiata* flower, we examined the flower’s anthesis speed, morphological processes at the cellular level, and conducted transcriptome analysis. These results provide essential knowledge for the conservation of this wild orchid species at risk of extinction.

### 3.1. Anthesis and Petal Withering

We have described the process of lip formation, anthesis, and petal wilting in the orchid *Habenaria radiata*. While many studies focus on flowering time, specifically the transition from the vegetative to reproductive phase [38], few studies address anthesis speed. *Habenaria radiata* takes 3.5 to 4 h for anthesis, and the flowers remain open for at least one week if they are not pollinated (Figure 1). However, we found that the removal of pollinia—clusters of pollen grains typically produced in orchids—caused rapid wilting of the petals (Figure 1). Our preliminary data suggest that pollination, rather than the removal of pollinia, triggers petal wilting. This implies that the flower possesses a mechanism to quickly wilt the petals that have completed their role in sensing pollination and attracting pollen vectors.

### 3.2. Lip Serration

A distinctive feature of this plant is lip serration, and we found that initial cell divisions form protrusions along the lip margin, and subsequently, cells in the serrated region elongate in the same direction, deepening the serrations. While it is known that *CUC* and *TCP* genes are involved in serration development in the leaves of *Arabidopsis thaliana* [29,30,31,32,33,35,36], the serrations on *Arabidopsis* leaves do not deepen. We found that these genes were expressed in floral buds, but other unknown factors may be involved in deepening the serration at a later stage. We found that directional cell growth is important for deepening the serration. Polarized cell growth in plants is regulated by the arrangement of cortical microtubules [39]; therefore, the relationship between the cytoskeleton and serration growth remains a topic for further investigation. To determine whether these genes are essential for lip development, research on mutants or transgenic plants is required. However, it has not been possible to generate these plants in *Habenaria radiata*. The establishment of cell culture and transformation methods will help to understand the molecular mechanisms of lip development.

Plants that form petal serrations are found in several plant families, including Caryophyllaceae, Celastraceae, Cucurbitaceae, Myrtaceae, Saxifragaceae, Tropaeolaceae, and Orchidaceae, suggesting that petal serrations evolved independently across the angiosperms. Notably, species of *Cucumis* can have extremely long fringes extending several centimeters [40]. Most species with deep serration on the petal margin are pollinated by nocturnal hawk moths. There are three possible roles for deep serrations in attracting pollinators: the first is a visual effect, where the flower shape becomes more conspicuous, particularly for nocturnal butterflies that need to recognize the flowers. Second, serration increases the surface area for fragrance emission from the petals. *Habenaria radiata* possesses several varieties with strong fragrance, suggesting that fragrance emission can attract pollinators at night. The third possibility is that serrations provide a scaffold for visiting insects. Hawk moths visiting *Habenaria radiata* flowers have been observed grasping the lip fringes with their forelegs while inserting their proboscis deep into the spur [27,28]. An experiment removing the serrations showed no significant effect on the number of moth visits, but did reduce the time spent on the flower and the seed production [27]. These findings suggest that the primary role of lip serrations in *Habenaria radiata* is likely as a scaffold, rather than a visual attractant. The role of serrations as a scaffold has also been investigated in *Mitella pauciflora* and its pollinator, fungus gnats, where serrations were found to be important for insects landing [41].

### 3.3. Regional Differences in Lip Morphology and Conservation

We previously showed that lip morphology in *Habenaria radiata* varied across different ecological regions [28]. The difference in lip shape may be due to the regulation of cell division and elongation at the lip periphery, which can be analyzed using the techniques presented in this work. It is interesting to investigate whether these regional differences affect pollination and seed production efficiency and whether the morphology of surrounding pollinators, such as proboscis length, varies among habitats. *Habenaria radiata* is a near-threatened species so successful pollination is crucial for maintaining genetic diversity. Therefore, fundamental knowledge of both *Habenaria radiata* and its pollinators is essential for ecosystem conservation.

### 3.4. Conclusions

The wild terrestrial orchid *Habenaria radiata* has a lip that resembles the shape of a white bird in flight and is threatened with extinction due to wetland loss. The main objective of this study was to understand the processes underlying lip development and flowering in this species, including anthesis and lip opening. Lip serration is a characteristic feature of several plant species, including *Habenaria radiata*, and plays an important role in attracting pollinators. We found that lip serration forms through an initial localized cell division, followed by directional cell elongation. Additionally, we identified floral homeotic genes, some of which were differently expressed in the lateral petals and the lip, suggesting that they regulate the development of these two types of petals. Regional variations in the lip morphology of *Habenaria radiata* have been documented [28], and the morphological and genetic insights provided in this study are expected to help clarify the mechanisms behind this diversity.

## 4. Materials and Methods

### 4.1. Plant Materials and Growth Conditions

*Habenaria radiata* “Aoba” was purchased from a gardening store in Nara, Japan. The plants were grown in cultivation soil (Nippi Soil No.1, Nihon Hiryo Co. Ltd., Fujioka, Japan) covered with wet sphagnum, in a glasshouse at Seika campus, Kyoto Prefectural University, Seika, Japan. A GX200 digital camera (RICOH, Tokyo, Japan) was used for interval shooting (time-lapse), and photos were taken every 10 min from June 24 to July 7 in 2015. The movies were generated using Image J (version 1.53a) and iMovie (version 10.4).

### 4.2. Confocal Microscopy

Lips from floral buds at different developmental stages were dissected and fixed in a solution of 10% acetic acid and 90% ethanol for more than 1 h. After the ethanol series (10 min in each concentration: 100%, 70%, 50%, and 30%), the lips were stained in the 50 µg/mL calcofluor solution (Fluorescent Brightener 28, Sigma-Aldrich, MO, USA) for more than 5 min. Staining was observed with a TCS SP8 confocal microscope (Leica, Wetzlar, Germany) with 405 nm excitation and detection from 410 to 600 nm. Images were captured using a photomultiplier and a Z-stack with a step size of 0.84.

### 4.3. Cell Shape Quantification

Cell shape analysis was performed as previously described [42], with a few modifications. The epidermal cell contours were manually traced from image printouts, which were then scanned with an LiDE220 (Cannon, Tokyo, Japan) as PNG files with a resolution of 200 dpi. Image J (version 1.53a) was used for cell shape analysis. The scanned images were converted to 8-bit, and the brightness threshold range was set to ”Auto” to segment the cell contours. The “Analyze Particles” function was used to measure the area and fit ellipse to the cells. The cell area and the angle of the major axis of the fitted ellipse were taken for 80 cells from each image. The major axis angle was corrected to ensure that the vertical bisector of an isosceles triangle, aligned with the serrated region, was at a 90-degree angle. For result display, the LPX plugin Lpx Measure (measure mode: blobMeasure) (https://lpixel.net/en/products/lpixel-imagej-plugins/, accessed on 3rd December, 2024) and pseudo-color from “Lookup Tables” were used.

### 4.4. Transcriptome Analysis

Three independent samples were used for RNA extraction and RNA sequencing. Total RNA was isolated from leaves and floral buds (3, 4, or 5 mm in length) using a modified protocol of the RNeasy Plant Mini Kit (QIAGEN, Hulsterweg, Netherland) [43]. Library preparation, RNA sequencing, sequence assembly, and homology searches were performed as described previously [44]. Total trinity transcripts and putative genes were 771,822 and 447,223, respectively, and median and average contig length were 351 and 657.16, respectively. The data have been deposited with links to BioProject accession number PRJDB19824 in the DDBJ BioProject database. Accession numbers for each dataset are shown in Appendix A. Genes expressed in flowers were selected based on a false discovery rate (FDR) < 0.01, a total read count > 1, and log_2_ fold change (log_2_FC) > 1 compared to leaves.

### 4.5. RT-PCR

Total RNA was isolated from leaves, and sepals, lateral petals, lips, and columns from floral buds with 7 mm length, as described above. cDNAs were generated with ReverTra Ace qPCR RT Master Mix with gDNA Remover (TOYOBO, Osaka, Japan). PCR was performed using KAPA Taq Extra DNA Polymerase (Nippon Genetics, Tokyo, Japan). Oligonucleotide primers and denature temperature for PCR are listed in Appendix A.

### 4.6. Transformation of Arabidopsis Thaliana Mutants

*HrAP2*, *HrAG*, and *HrAGL6* genes were cloned into the qCR4-TOPO vector (Invitrogen, MA, USA) and subcloned into the pAN19 vector carrying CaMV 35S promoter and NOS terminator. The *35Spro:HrAP2, HrAG2, or HrAG3:NOSter* cassette was subcloned into the pBIN30 binary vector and transformed to *Agrobacterium tumefaciens* (*Rhizobium radiobacter*) strain GV3101_pMP90. The *35S:HrAP2* and *35S:HrAG2* constructs were then transformed into *ap2-3/+* and *ag-1/+* heterozygous plants, respectively, with the floral dip method [45]. T1 plants were selected on MS media containing 10 µg/mL glufosinate and genotyped for the *Arabidopsis AP2* and *AG* genes. The *HrAP2*, *HrAG3*, and *HrAGL6-C3* sequences have been deposited in the GenBank database with accession numbers LC856603, LC856604, and LC856605, respectively.

## Figures and Tables

**Figure 1 plants-14-00393-f001:**
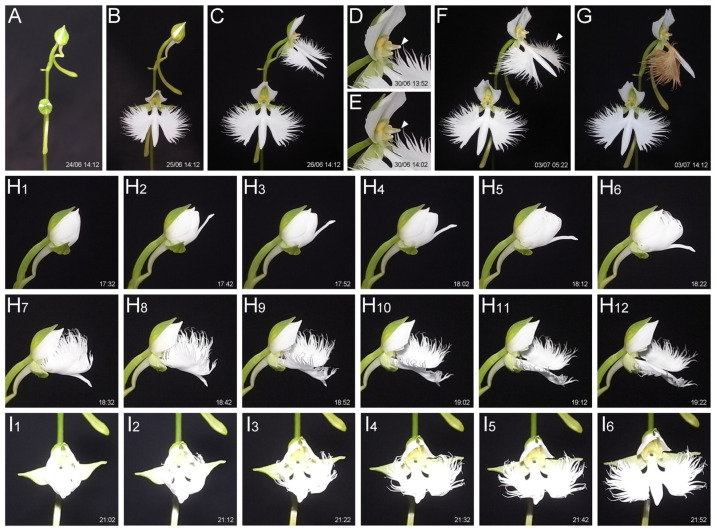
Flowering and withering of *Habenaria radiata* flowers captured by interval shooting (time-lapse). (**A**–**G**) Flowering and withering of flowers. The date (dd/mm) and time of capture are shown. Note that the upper flower, which bloomed later, withered earlier than the lower flower (**G**), probably due to the loss of pollinium (arrowheads in (**D**,**E**)) and subsequent pollination. (**H_1_**–**H_12_**,**I_1_**–**I_6_**) Side (**H_1_**–**H_12_**) and front (**I_1_**–**I_6_**) views of lip unfolding. The captured time is shown in each panel.

**Figure 2 plants-14-00393-f002:**
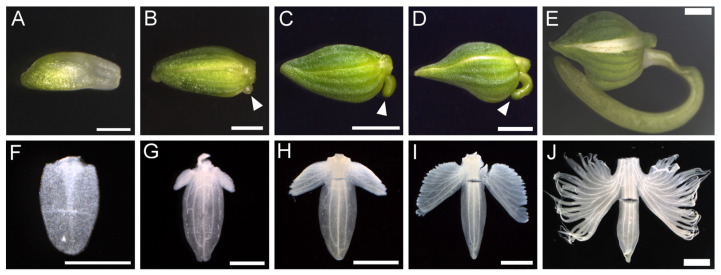
Lip development. (**A**–**E**) Floral buds at different stages. Arrowheads in B, C, and D indicate the growing spur. (**F**–**J**) Lip inside the floral bud shown in (**A**–**E**). The lengths of the floral buds are 1 mm (**A**,**F**), 2 mm (**B**,**G**), 4 mm (**C**,**H**), 6 mm (**D**,**I**), and 7 mm (**E**,**J**). Scale bars: A, F = 0.5 mm; B, G = 1 mm; C, D, E, H, I, J = 2 mm.

**Figure 3 plants-14-00393-f003:**
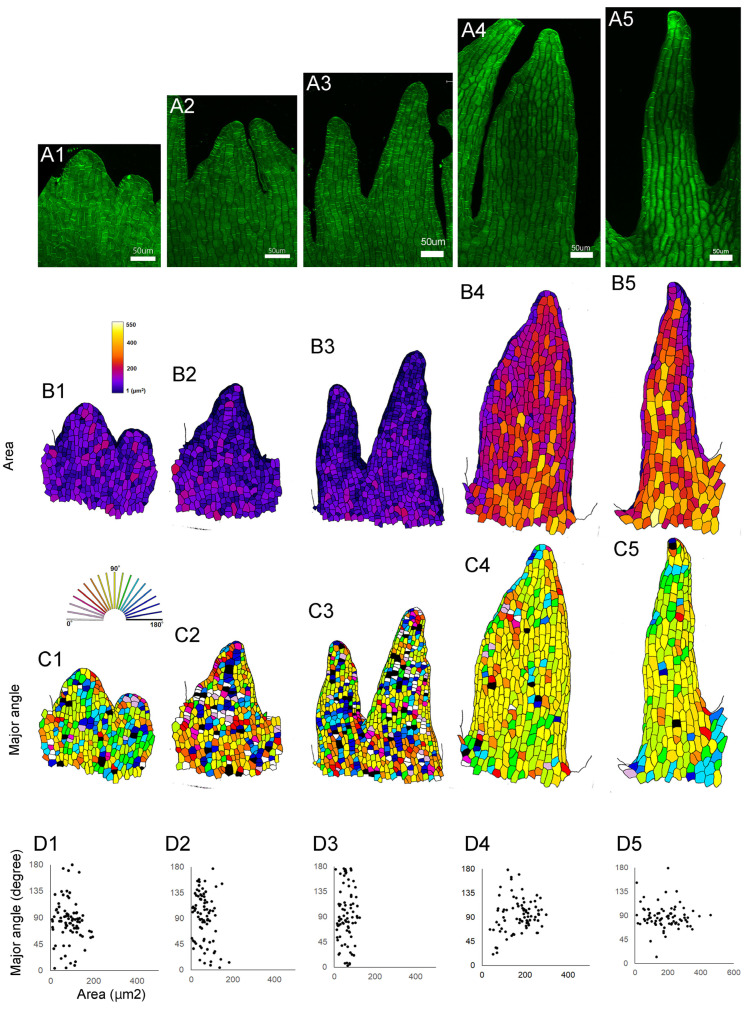
Cell shape changes during the development of lip serration. (**A**) Confocal laser microscopy images of petal margin cells. Petals from early to late (**A1**–**A5**) stages were excised, stained, and observed. (**B1**–**B5**) Distribution of cell area. (**C1**–**C5**) Elongation direction of each cell. The direction of serration elongation was set as 90 degrees, with the angles relative to this direction shown in different colors. (**D1**–**D5**) Scatter plots of cell area and elongation direction for each stage. Up to time point 3, cell proliferation occurs, and from time point 4 onward, the serrations deepen due to polarized cell elongation. Scale bars: 50 µm.

**Figure 4 plants-14-00393-f004:**
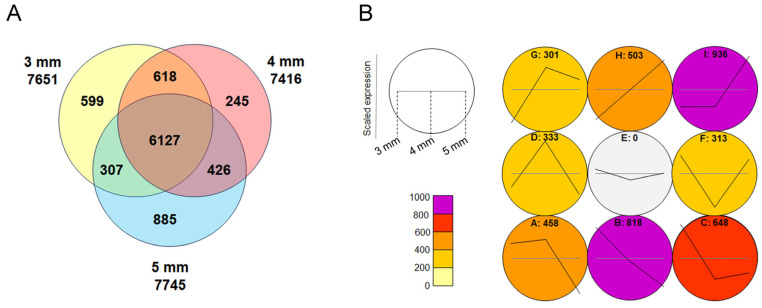
Transcriptome analysis of floral buds. (**A**) Venn diagram showing genes expressed more than twice as much in floral buds of 3 mm, 4 mm, and 5 mm sizes compared to leaves. (**B**) Self-organizing map (SOM) analysis of the genes expressed in floral buds. The letters represent clusters with similar expression patterns, and the numbers indicate the gene number in each cluster. Clusters G, H, and I show an increase in expression during bud development, while clusters A, B, and C show a decrease in expression.

**Figure 5 plants-14-00393-f005:**
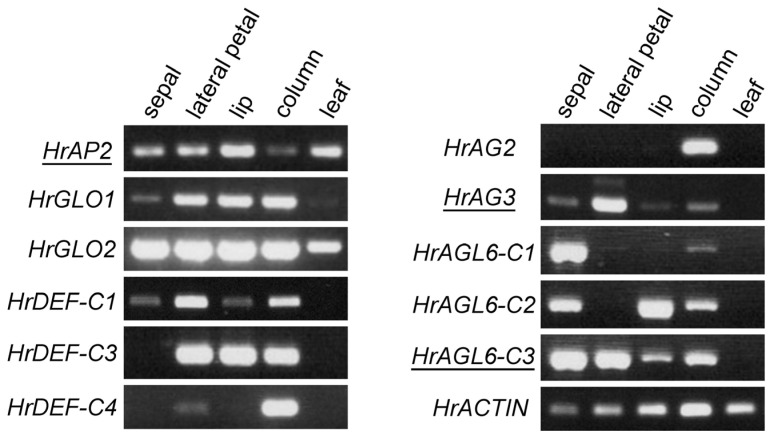
RT-PCR of floral homeotic and MADS genes in *Habenaria radiata*. Underlined genes are reported for the first time in this work. HrACTIN was used as the control.

**Figure 6 plants-14-00393-f006:**
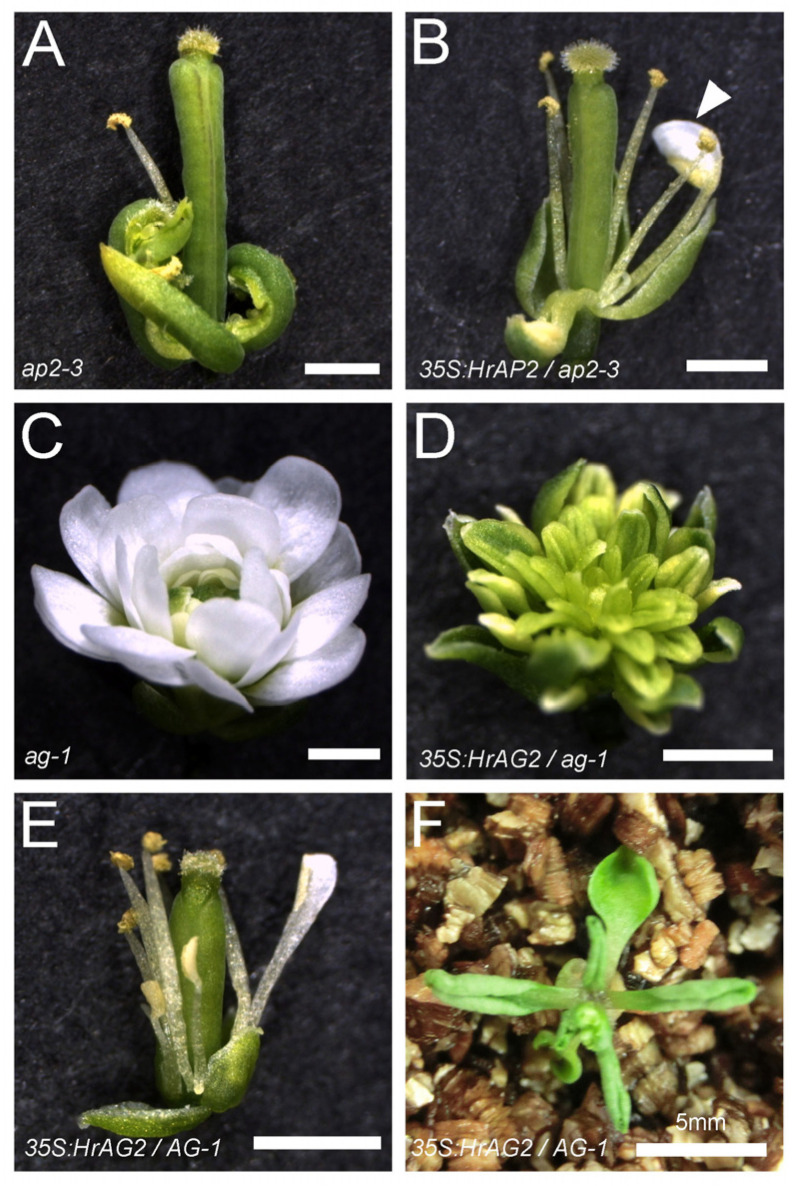
Overexpression of *HrAP2* and *HrAG2* in *Arabidopsis thaliana*. (**A**) *ap2-3* mutant. (**B**) *35S:HrAP2* plant in the *ap2-3* background. Arrowhead indicates the petaloid stamen. Note that sepal-like organs are generated in the first whorl, and more stamens were produced compared to the *ap2-3* mutant. (**C**) *ag-1* flower. (**D**) *35S:HrAG2* plant in the *ag-1* background. Note that stamen-like organs are generated. (**E**,**F**) *35S:HrAG2* plants in *AG-1* (wild-type sibling) background. (**E**) Stamens are generated instead of petals in the second whorl. (**F**) Vegetative phenotype showing hyponastic growth in leaves, resulting in the curled leaves. Scale bars: A, B, C, D, E = 1 mm; F = 5 mm.

**Figure 7 plants-14-00393-f007:**
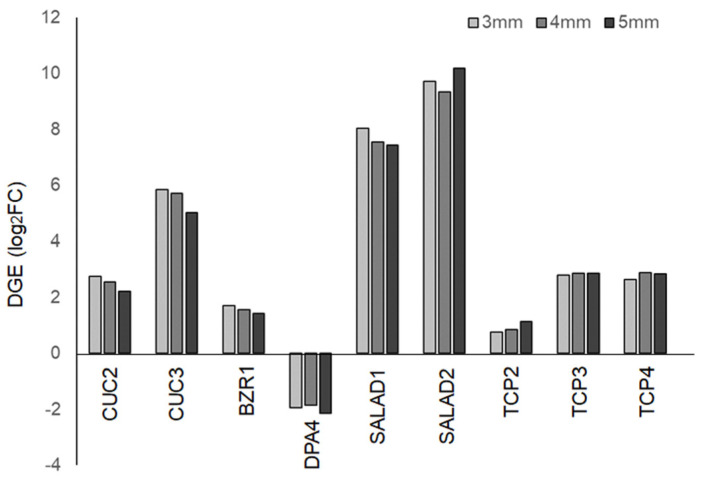
Digital gene expression (DGE) of genes involved in serration formation in floral buds.

## Data Availability

All data are contained within the article.

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
