# Peer review of "Morphological and Transcriptome Analysis of the Near-Threatened Orchid *Habenaria radiata* with Petals Shaped Like a Flying White Bird"

_plants, 2025, doi:10.3390/plants14030393_

Round 1
Reviewer 1 Report
Comments and Suggestions for Authors
The manuscript is in line with the scope of Plants and deals with very important scientific issues. The quality of the manuscript is good and it has many strengths, but I would like to mention the most important weaknesses and points to be corrected.
1. I recommend changing the keywords to avoid repeating all the terms already mentioned in the manuscript title.
2. I would advise the authors to add at the end of the introduction (lines 90-95) the questions they sought to answer in their research. These questions would be of great help to each reader in finding the information relevant to them.
3. Internet sources (lines 71, 76, etc.) should be cited as normal literature sources, as indicated in the requirements for citing literature sources.
4. I think that the format of Figure 3 is too small. The graphs are so small that it is not possible to see what they represent, and the scale bars are not visible. I would advise that the illustration be enlarged significantly so that it is not only clear when the individual elements are enlarged, but also so that the entire picture can be seen and understood clearly.
5. The authors' decision to move the Materials and Methods section to the end of the manuscript is completely unjustified. As a result, the Results section contains a large number of parts specific to the Materials and Methods section that would not be needed if this section were placed after the Introduction. For example, such information relevant to the Materials and Methods section is contained in lines 96-101, 122-124, 138-140 and many other places in the Results section. I strongly recommend that the manuscript be reorganised in a logical sequence.
6. In my opinion, the discussion is too weak. Therefore (as I suggested in comment 2), the discussion should be rewritten according to the questions raised and answered in the study. The last paragraph of the discussion should be based on more literature sources on the pollination and conservation needs of Orchidaceae. In addition, I doubt very much whether knowledge of the species helps to protect ecosystems. Maybe it is the other way around, that knowledge of the needs of the species helps to organise the protection of the species itself, its habitats and ecosystems.
7. I recommend including a description of the statistical analysis methods.
8. As the study covers a large number of issues, I strongly recommend that a conclusion or summary section be prepared to provide a focused and clear answer to the questions (again related to comment 2) to which the study has sought answers. This would add to the value of the paper.
Author Response
Comment 1. I recommend changing the keywords to avoid repeating all the terms already mentioned in the manuscript title.
Response 1: Thank you for this comment, we have updated the keywords.
Comment 2. I would advise the authors to add at the end of the introduction (lines 90-95) the questions they sought to answer in their research. These questions would be of great help to each reader in finding the information relevant to them.
Response 2: Our main questions are presented in lines 67-69 and 87-89 (the mechanism by which the lip develops into its unique shape, how the petals become serrated at their margins, and how the lip opens during anthesis). To avoid repetition, we have outlined what this manuscript has done to address these questions in the final paragraph of the Introduction.
Comment 3. Internet sources (lines 71, 76, etc.) should be cited as normal literature sources, as indicated in the requirements for citing literature sources.
Response 3. Thank you for pointing that out. We have included the internet sources in the literature references.
Comment 4. I think that the format of Figure 3 is too small. The graphs are so small that it is not possible to see what they represent, and the scale bars are not visible. I would advise that the illustration be enlarged significantly so that it is not only clear when the individual elements are enlarged, but also so that the entire picture can be seen and understood clearly.
Response 4. This was due to the layout of the paper; we wanted to avoid creating large empty spaces in the manuscript. In the revised version, Figure 3 has been enlarged so that readers can more easily recognize the figure panels.
Comment 5. The authors' decision to move the Materials and Methods section to the end of the manuscript is completely unjustified. As a result, the Results section contains a large number of parts specific to the Materials and Methods section that would not be needed if this section were placed after the Introduction. For example, such information relevant to the Materials and Methods section is contained in lines 96-101, 122-124, 138-140 and many other places in the Results section. I strongly recommend that the manuscript be reorganised in a logical sequence.
Response 5. We followed the manuscript preparation guidelines, and the 'Plants' format places the Materials and Methods (M&M) section after the Results. Unfortunately, we cannot change this format. We recommend that readers refer to the M&M section if they have any questions while reading the Results. We prefer not to mix the information from the Results and M&M sections.
Comment 6. In my opinion, the discussion is too weak. Therefore (as I suggested in comment 2), the discussion should be rewritten according to the questions raised and answered in the study. The last paragraph of the discussion should be based on more literature sources on the pollination and conservation needs of Orchidaceae. In addition, I doubt very much whether knowledge of the species helps to protect ecosystems. Maybe it is the other way around, that knowledge of the needs of the species helps to organise the protection of the species itself, its habitats and ecosystems.
Response 6. Thank you for this important comment. As mentioned in response 2, we outlined our main questions in the Introduction and provided answers to these questions in the Discussion. Specifically, we focused on serration formation to understand the mechanism behind the variation in petal shape. In the last paragraph, we discussed the pollination of Habenaria radiata. However, there are few reports on this topic for Habenaria radiata and related species, which is why we concentrated on this species to understand the developmental basis of flower formation, especially as it is a near-threatened species. We believe that our results, along with our previous work (Tachibana et al., 2020), provide valuable ecological, cellular, and genetic insights into the basics of this species, which can contribute to its conservation. Since these plants grow in wetlands, which are disappearing due to human development, protecting this species also helps safeguard the wetland ecosystems that support many organisms.
Comment 7. I recommend including a description of the statistical analysis methods.
Response 7. We included this in the M&M section by citing our previous work, to avoid repetition of the detail description.
Comment 8. As the study covers a large number of issues, I strongly recommend that a conclusion or summary section be prepared to provide a focused and clear answer to the questions (again related to comment 2) to which the study has sought answers. This would add to the value of the paper.
Response 8. Thank you for this comment. We believe we have addressed it in Responses 2 and 6.
Reviewer 2 Report
Comments and Suggestions for Authors
Major:
This paper is interesting and publishable, but the authors must improve their logic and discussion. Unfortunately, it implies the major revision of the text. In particular, they must directly (even briefly) discuss the role of all investigated genes (lines 213-248 and 272) in the development of the flower of Habenaria radiata in general and the flower lip in particular. The authors must articulate this explicitly if neither of these genes is essential in lip development. Is it true that only CUC and BZR genes are involved in lip formation in H. radiata? 'These factors (genes CUC and BZR?) are suggested to be involved in lip serration formation' (lines 255-256). Who made this suggestion? 'We found that these genes were expressed in floral buds, but other unknown factors may be involved in deepening the serration at a later stage' (lines 281-282). This proposition is exciting, but the lack of proper discussion makes it unclear to the reader why the authors discussed unknown factors in development when the role of numerous genes was never adequately discussed."
Minor:
"... the floral ABCE model" (lines 43-44). Scientists typically discuss the ABC and quartet models. A brief, well-focused review of both models, their limitations, and their role (actual or potential) in developing orchid flowers is still necessary (either in the introduction or discussion). Currently, the reader can only find relevant semantic fragments in the text. If neither of these models helps explain lip development, the authors must stress this.
Author Response
Comment 1 (Major): This paper is interesting and publishable, but the authors must improve their logic and discussion. Unfortunately, it implies the major revision of the text. In particular, they must directly (even briefly) discuss the role of all investigated genes (lines 213-248 and 272) in the development of the flower of Habenaria radiata in general and the flower lip in particular. The authors must articulate this explicitly if neither of these genes is essential in lip development. Is it true that only CUC and BZR genes are involved in lip formation in H. radiata? 'These factors (genes CUC and BZR?) are suggested to be involved in lip serration formation' (lines 255-256). Who made this suggestion? 'We found that these genes were expressed in floral buds, but other unknown factors may be involved in deepening the serration at a later stage' (lines 281-282). This proposition is exciting, but the lack of proper discussion makes it unclear to the reader why the authors discussed unknown factors in development when the role of numerous genes was never adequately discussed."
Response 1: Thank you for these important comments. The reviewer suggests adding information about the direct role of genes in Habenaria radiata. To address this, gene knock-out or overexpression studies are required. However, we have not been successful in generating such plants so far. We attempted callus generation from this plant to create transgenic plants, but we were unable to obtain good callus (the cells regenerated initially but soon turned black and died). Therefore, it is currently impossible to demonstrate that these genes are essential for petal development in Habenaria radiata—a goal we hope to achieve in the future. We have included the following sentence in the Discussion section: 'To determine whether these genes are essential for lip development, research on mutants or transgenic plants is required. However, it has not been possible to generate these plants in Habenaria radiata. The establishment of cell culture and transformation methods will help to understand the molecular mechanisms of lip development.
We did not claim that CUC and BZR are the 'only' genes involved in lip serration in Habenaria radiata; we acknowledge that many other genes are likely involved in this process. As mentioned in the manuscript, we focused on 'serration genes' from leaves in Arabidopsis and tomato, as these genes have been identified in these organs. We examined these differentially expressed genes (DEGs) in our transcriptome analysis of H. radiata flowers. Again, we are unable to show the 'direct involvement' of these genes in lip serration due to the absence of mutants or transgenic plants. Thus, we can only 'suggest' the involvement of these genes in lip development. Generating mutants and transgenic plants to confirm our hypotheses is one of our key goals for future research.
Comment 2 (Minor): ... the floral ABCE model" (lines 43-44). Scientists typically discuss the ABC and quartet models. A brief, well-focused review of both models, their limitations, and their role (actual or potential) in developing orchid flowers is still necessary (either in the introduction or discussion). Currently, the reader can only find relevant semantic fragments in the text. If neither of these models helps explain lip development, the authors must stress this.
Response 2: There are many reports on the floral ABCE model in orchids, including Habenaria radiata, as mentioned in the Introduction with references. We have added the phrase 'almost all flowering plants' in the Introduction. To avoid overly detailed descriptions of less relevant topics, we believe it would be more helpful for readers to consult the references as needed.
Round 2
Reviewer 1 Report
Comments and Suggestions for Authors
The manuscript was revised after the first round of reviews, but very minimally and many of the comments were not taken into account. For example, comment 2 was about research questions. They were not in the article and are not there. There is not even a clear research objective stated. The aim and objectives, or the research questions to which the answers were sought, should be stated, and the conclusions should provide clear and concise answers to the questions raised.
Statistical methods are not mentioned. Each article is based on different data sets and different methods of statistical analysis are used. How can the distribution of the data be assessed in the next article? Which datasets are the graphs in Figure 3 D1-D5 based on? This is essential information.
The discussion is also largely unfinished and very short.
There are no clearly formulated conclusions.
As the methods are at the end of the paper, the results section remained rather vague. How can one assess the results without being familiar with the methodology? I am very glad that no one has yet thought to delete the methods section from articles completely. I would not be surprised if it will soon be necessary to look for the methods in Supplementary Materials or other databases for each article.
Author Response
Comments of 1st reviewer:
The manuscript was revised after the first round of reviews, but very minimally and many of the comments were not taken into account. For example, comment 2 was about research questions. They were not in the article and are not there. There is not even a clear research objective stated. The aim and objectives, or the research questions to which the answers were sought, should be stated, and the conclusions should provide clear and concise answers to the questions raised.
Statistical methods are not mentioned. Each article is based on different data sets and different methods of statistical analysis are used. How can the distribution of the data be assessed in the next article? Which datasets are the graphs in Figure 3 D1-D5 based on? This is essential information.
The discussion is also largely unfinished and very short.
There are no clearly formulated conclusions.
As the methods are at the end of the paper, the results section remained rather vague. How can one assess the results without being familiar with the methodology? I am very glad that no one has yet thought to delete the methods section from articles completely. I would not be surprised if it will soon be necessary to look for the methods in Supplementary Materials or other databases for each article.
Response to 1st reviewer:
Thank you for the second-round review. We have carefully reviewed the comments and made revisions as thoroughly as possible.
First, we have clearly stated the objective of this research in the last paragraph of the Introduction. Our objective is to understand the morphology and developmental process of the Habenaria radiata flower, with a particular focus on the lip, which plays a critical role in attracting pollinators.
Figure 2D consists of scatter plots for each stage shown in Figures 2A-C, so no statistical analysis is necessary. These panels illustrate the process of cellular shape changes during lip generation. To avoid confusion, we have added an explanation to the figure legend.
We have added a description of the flowering and wilting processes to the Discussion section and divided it into several paragraphs for clearer presentation.
For the Materials and Methods section, we have added a brief explanation of the methods in the Results section. For detailed methodology, please refer to the Materials and Methods section.
Reviewer 2 Report
Comments and Suggestions for Authors
I do not see much improvement in the text, but I see the reasons for it based on the author's detailed responses.
Author Response
Comment of 2nd reviewer:
I do not see much improvement in the text, but I see the reasons for it based on the author's detailed responses.
Response:
Thank you for reviewing our manuscript.
Round 3
Reviewer 1 Report
Comments and Suggestions for Authors
The authors of the manuscript have taken on board some of the comments made in previous reviews, or have provided reasoned responses.
I fully understand and agree that authors can have strong opinions and not accept the reviewers' comments, but there would be less misunderstanding if they responded immediately to the comments made and clearly expressed their opinions and preferences.
Nevertheless, one significant comment was not taken into account by the authors during the two rounds of peer review: there is still a lack of a clear and concise summary of the results in the form of a conclusion or concluding paragraph.
Author Response
Comments of reviewer1:
The authors of the manuscript have taken on board some of the comments made in previous reviews, or have provided reasoned responses.
I fully understand and agree that authors can have strong opinions and not accept the reviewers' comments, but there would be less misunderstanding if they responded immediately to the comments made and clearly expressed their opinions and preferences.
Nevertheless, one significant comment was not taken into account by the authors during the two rounds of peer review: there is still a lack of a clear and concise summary of the results in the form of a conclusion or concluding paragraph.
Response:
We have added a conclusion paragraph following the suggestion.